# Modern Neural Networks Generalize on Small Data Sets

**Matthew Olson**
Department of Statistics
Wharton School University of Pennsylvania
Philadelphia, PA 19104
maolson@wharton.upenn.edu

**Abraham J. Wyner**
Department of Statistics
Wharton School University of Pennsylvania
Philadelphia, PA 19104
ajw@wharton.upenn.edu

**Richard Berk**
Department of Statistics
Wharton School University of Pennsylvania
Philadelphia, PA 19104
berkr@wharton.upenn.edu

## Abstract

In this paper, we use a linear program to empirically decompose fitted neural networks into ensembles of low-bias sub-networks. We show that these sub-networks are relatively uncorrelated which leads to an internal regularization process, very much like a random forest, which can explain why a neural network is surprisingly resistant to overfitting. We then demonstrate this in practice by applying large neural networks, with hundreds of parameters per training observation, to a collection of 116 real-world data sets from the UCI Machine Learning Repository. This collection of data sets contains a much smaller number of training examples than the types of image classification tasks generally studied in the deep learning literature, as well as non-trivial label noise. We show that even in this setting deep neural nets are capable of achieving superior classification accuracy without overfitting.

## 1   Introduction

A recent focus in the deep learning community has been resolving the "paradox" that extremely large, high capacity neural networks are able to simultaneously memorize training data and achieve good generalization error. In a number of experiments, Zhang et al. [24] demonstrated that large neural networks were capable of both achieving state of the art performance on image classification tasks, as well as perfectly fitting training data with permuted labels. The apparent consequence of these observations was to question traditional measures of complexity considered in statistical learning theory.

A great deal of recent research has aimed to explain the generalization ability of very high capacity neural networks [14]. A number of different streams have emerged in this literature [18, 16, 17]. The authors in Zhang et al. [24] suggest that stochastic gradient descent (SGD) may provide implicit regularization by encouraging low complexity solutions to the neural network optimization problem. As an analogy, they point out that SGD applied to under-determined least squares problems produces solutions with minimal $\ell_2$ norm. Other streams of research have aimed at exploring the effect of margins on generalization error [1, 16]. This line of thought is similar to the margin-based views of AdaBoost in the boosting literature that bound test performance in terms of the classifier's confidence

in its predictions. Other research has investigated the sharpness of local minima found by training a neural network with SGD [9, 18]. The literature is extensive, and this review is far from complete.

The empirical investigations found in this literature tend to be concentrated on a small set of image classification data sets. For instance, every research article mentioned in the last section with an empirical component considers at least on of the following four data sets: MNIST, CIFAR-10, CIFAR-100, or ImageNet. In fact, in both NIPS 2017 and ICML 2017, over half of all accepted papers that mentioned "neural networks" in the abstract or title used one of these data sets. All of these data sets share characteristics that may narrow their generality: similar problem domain, very low noise rates, balanced classes, and relatively large training sizes (60k training points at minimum).

In this work, we consider a much richer class of *small* data sets from the UCI Machine Learning Repository in order to study the "generalization paradox." These data sets contain features not found in the image classification data, such as small sample sizes and nontrivial, heteroscedastic label noise. Although not without its faults [20], the UCI repository provides a much needed alternative to the standard image data sets.

As part of our investigation we will establish that large neural networks with tens of thousands of parameters are capable of achieving superior test accuracy on data sets with only hundreds of observations. This is surprising, as it is commonly thought that deep neural networks require large data sets to train properly [19, 7]. We believe that this gap in knowledge has led to the common misbelief that unregularized, deep neural networks will necessarily overfit the types of data considered by small-data professions. In fact, we establish that with minimal tuning, deep neural networks are able to achieve performance on par with a random forest classifier, which is considered to have state-of-the-art performance on data sets from the UCI Repository [10].

The mechanism by which a random forest is able to generalize well on small data sets is straightforward: a random forest is an ensemble of low-bias, decorrelated trees. Randomization combined with averaging reduces the ensemble's variance, smoothing out the predictions from fully grown trees. It is clear that a neural network should excel at bias reduction, but the way in which it achieves variance reduction is much more mysterious. The same paradox has been examined at length in the literature on AdaBoost, and in particular, it has been conjectured that the later stages of AdaBoost serve as a *bagging* phase which reduces variance [4, 5, 6, 23].

One of the central aims of this paper is to identify the variance stabilization that occurs when training a deep neural network. To this end, the later half of this paper focuses on empirically decomposing a neural network into an ensemble of sub-networks, each of which achieves low training error and less than perfect pairwise correlation. In this way, we view neural networks in a similar spirit to random forests. One can use this perspective as a window to viewing the success of recent residual network architectures that are fit with hundreds or thousands of hidden layers [13, 22]. These deeper layers might serve more as a bagging mechanism, rather than additional levels of feature hierarchy, as is commonly cited for the success of deep networks [3].

The key takeaways from this paper are summarized as follows:

- Large neural networks with hundreds of parameters per training observation are able to generalize well on small, noisy data sets.

- Despite a bewildering set of tuning parameters [2], neural networks can be trained on small data sets with minimal tuning.

- Neural networks have a natural interpretation as an ensemble of low-bias classifiers whose pairwise correlations are less than one. This ensemble view offers a novel perspective on the *generalization paradox* found in the literature.

## 2   Ensemble View of Deep Neural Networks

In this section, we establish that a neural network has a natural interpretation as an ensemble classifier. This perspective allows us to borrow insights from random forests and model stacking to gain better insight as to how a neural network with many parameters per observation is still able to generalize well on small data sets. We also outline a procedure for decomposing fitted neural networks into ensemble components of low bias, decorrelated sub-networks. This procedure will be illustrated for a number of neural networks fit to real data in Section 3.3.

## 2.1 Network Decomposition

We will begin by recalling some familiar notation for a feed-forward neural network in a binary classification setting. In the case of a network with $L$ hidden layers, each layer with $M$ hidden nodes, we may write the network's prediction at a point $x \in \mathbb{R}^p$ as

$$z^{\ell+1} = W^{\ell+1} g(z^\ell) \ \ \ell = 0, \ldots, L$$
$$f(x) = \sigma(z^{L+1}) \tag{1}$$

where $\sigma$ is the sigmoid function, $g$ is an activation function, $W^{L+1} \in \mathbb{R}^{1 \times M}$, $W^1 \in \mathbb{R}^{M \times p}$, and $W^\ell \in \mathbb{R}^{M \times M}$ for $\ell = 2, \ldots, L$ (and $z^0 \equiv x$). Assume that any bias terms have been absorbed. For the models considered in this paper, $L = 10$, $M = 100$, and $g$ is taken to be the *elu* activation function [8]. It is also helpful to abuse notation a bit, and to write $z^\ell(x)$ as the output at hidden layer $\ell$ when $x$ is fed through the network.

There are many ways to decompose a neural network into an ensemble of sub-networks: one way to do this is at the final hidden layer. Let us fix an integer $K \leq M$ and consider a matrix $\alpha \in \mathbb{R}^{M \times K}$ such that $\sum_{k=1}^{K} \alpha_{m,k} = W_{1,m}^{L+1}$ for $m = 1, \ldots, M$. We can then write the final *logit* output as a combination of units from the final hidden layer:

$$
\begin{aligned}
z^{L+1}(x) &= W^{L+1} g(z^L(x)) \\
&= \sum_{m=1}^{M} W_{1,m}^{L+1} g(z_m^L(x)) \\
&= \sum_{m=1}^{M} \sum_{k=1}^{K} \alpha_{m,k} g(z_m^L(x)) \\
&= \sum_{k=1}^{K} \sum_{m=1}^{M} \alpha_{m,k} g(z_m^L(x)) \\
&= \sum_{k=1}^{K} f_k(x)
\end{aligned}
\tag{2}
$$

where $f_k(x) = \sum_{m=1}^{M} \alpha_{m,k} g(z_m^L(x))$. In words, we have simply decomposed the final layer of the network (at the *logit level*) into a sum of component networks. The weights that define the $k^{th}$ sub-network are stored in the $k^{th}$ column of $\alpha$.

We will find in Section 3 that networks trained on a number of binary classification problems have decompositions such that each $f_k$ fits the training data perfectly, and such that out-of-sample correlation between each $f_i$ and $f_j$ is relatively small. This situation is reminiscent of how a random forest works: by averaging the outputs of low-bias, low-correlation trees. We argue that it is through this implicit regularization mechanism that overparametrized neural networks are able to simultaneously achieve zero training error and good generalization performance.

## 2.2 Ensemble Hunting

The decomposition in Equation 2 is of course entirely open-ended without further restrictions on $\alpha$, the weights defining each sub-network. Broadly speaking, we want to search for a set of ensemble components that are both diverse and low bias. As a proxy for the latter, we impose the constraint that each sub-network achieves very high training accuracy. We will restrict our analysis to networks that obtain 100% training accuracy, and we will demand that each sub-network $f_k$ do so as well.

As a proxy for diversity, we desire that each sub-network in the ensemble should be built from a different part of the full network, as much to the extent that is possible. One strategy for accomplishing this is to require that the columns of $\alpha$ are sparse and have non-overlapping components. In practice, we found that the integer programs required to find these matrices were computationally intractable when coupled with the other constraints we consider. Our approach was simply to force a random selection of half the entries of each column to be zero through linear constraints.

We will now outline our ensemble search more precisely. For each of the $K$ columns of $\alpha$, we sampled integers $(m_{1,k}, \ldots, m_{M/2,k})$ uniformly without replacement from the integers 1 to $M$, and

we included the linear constraints $\alpha_{m_{j,k},k} = 0$. Thus, we constrained each sub-network $f_k$ to be a weighted sum of no more than $M/2$ units from the final hidden layer. Under this scheme, two sub-networks share $0.25M$ hidden units on average, and any given hidden unit appears in about half of the sub-networks. We then used linear programming to find a matrix $\alpha \in \mathbb{R}^{M \times K}$ that satisfied the required constraints:

$$\sum_{k=1}^{K} \alpha_{m,k} = W_{1,m}^{L+1}, \qquad\qquad\qquad\qquad 1 \leq m \leq M$$

$$\alpha_{m_{j,k},k} = 0, \qquad\qquad\qquad\qquad 1 \leq j \leq M/2, \quad 1 \leq k \leq K$$

$$\left( \sum_{m=1}^{M} \alpha_{m,k} g(z_m^L(x_i)) \right) y_i \geq 0, \qquad\qquad 1 \leq i \leq n, \ \ 1 \leq k \leq K$$

In summary, the first set of constraints ensures that the sub-networks $f_k$ decompose the full network, that is, so that $z^{L+1}(x) = \sum_{k=1}^{K} f_k(x)$ for all $x \in \mathbb{R}^p$. The second set of constraints zeros out half entries for each column of $\alpha$, encouraging diversity among the sub-networks. The final set of constraints ensures that each sub-network achieves 100% accuracy on the training data (non-negative margin).

Notice that we are simply looking for any feasible $\alpha$ for this system, and these constraints are rough heuristics that our sub-networks are diverse and low-bias. We could have additionally incorporated a loss function which further penalized similarity among the columns of $\alpha$, such as maximizing pairwise distances between elements. However, most reasonable distance measures - such as norms or Kullback-Leibler divergence - are convex, and maximizing a convex function is difficult. Finally, we emphasize that these constraints are very demanding, and rule out trivial decompositions. For instance, in a number of experiments, we were not able to find any feasible $\alpha$ for networks with a small number of hidden layers.

## 2.3 Model Example

As a first application of *ensemble hunting*, we will consider a simulated model in two dimensions. The response $y$ takes values $-1$ and $1$ according to the following probability model, where $x \in [-1,1]^2$:

$$p(y = 1|x) = \begin{cases} 1 & \text{if } \|x\|_2 \leq 0.3 \\ 0.15 & \text{otherwise .} \end{cases}$$

The Bayes rule for this model is to assign a label $y = 1$ if $x$ is inside a circle centered at the origin with radius $0.3$, and to assign $y = -1$ otherwise. This rule implies a minimal possible classification error rate of approximately $10\%$.

Our training set consists of $400$ $(x, y)$ pairs, where the predictors $x$ form an evenly spaced grid of values on $[-1,1]^2$. We train two classifiers: a neural network with $L = 10$ hidden layers and $M = 100$ hidden nodes, and a random forest. In each case, both classifiers are trained until they achieve 100% training accuracy. The decision surfaces implied by these classifiers, as well as the Bayes rule, are plotted in Figure 1. Evaluated on a hold-out set of size $n = 10,000$, the neural network and random forest achieve test accuracies of $85\%$ and $84\%$, respectively. Inspecting Figure 1, we see that although each classifier fits the training data perfectly, the fits around the *noise points* outside the circle are confined to small neighborhoods. Our goal is to explain how these types of fits occur in practice.

The bottom two figures in Figure 1 show the decision surfaces produced by the random forest and a single random forest tree. Comparing these surfaces illustrates the power of ensembling: the single tree has smaller accuracy than the large forest, as evidenced by the black patches outside of the circle. However, these patches are relatively small, and when all the trees are averaged together, much of these get smoothed out to the Bayes rule. Averaging works because the random forest trees are diverse by construction. We would like to extend this line of reasoning to explain the fit produced by the neural network.

Unlike a random forest, for which it is relatively easy to find sub-ensembles with low training error and low correlation, the corresponding search in the case of neural networks requires more work.

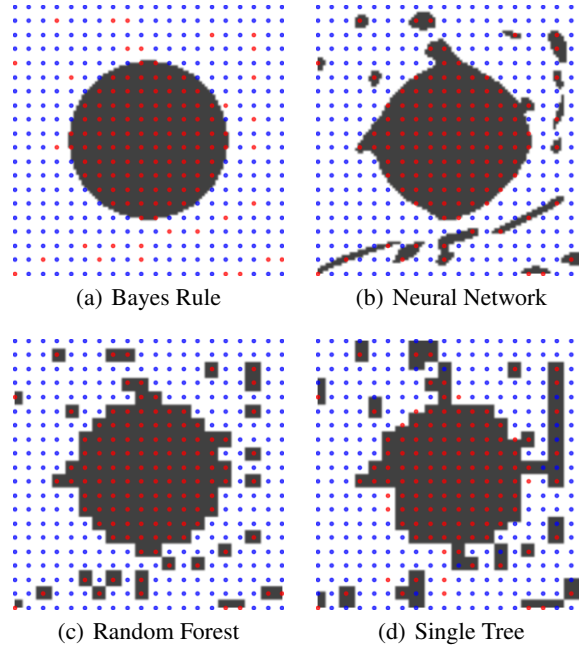

|  |  |
|---|---|
| (a) Bayes Rule | (b) Neural Network |
| (c) Random Forest | (d) Single Tree |

Figure 1: In each figure, the black region indicates points for which the classifier returns a label of $y = 1$. Training points with class label $y = 1$ are shown in red, and points with class label $y = -1$ are shown in blue.

Using the ensemble-hunting procedure outlined in the previous section, we decompose the network into $K = 8$ sub-networks $f_1, \ldots, f_8$, and we plot their associated response surfaces in Figure 2. The test accuracies of these sub-networks range from $79\%$ to $82\%$, and every classifier fits the training data perfectly by construction. When examining these surfaces, it is curious that they all look somewhat different, especially near the edges of the domain. Using the test set, we compute that the average error correlation across sub-networks is $60\%$. We would like to emphasize that in this example, the performance of both classifiers is actually quite good, especially compared to a more simple procedure such as CART.

One surprising conclusion from this exercise was that the final layer of our fitted neural network was highly redundant: the final layer could be decomposed as 8 distinct classifiers, each of which achieved 100% training accuracy. Common intuition for the success of neural networks suggests that deep layers provide a rich hierarchy of features useful for classification [3]. For instance, when training a network to distinguish cats and dogs, earlier layers might be able to detect edges, while later layers learn to detect ears or other complicated shapes. There are no complicated features to learn in this example: the Bayes decision boundary is a circle, which can be easily constructed from a network with one hidden layer and a handful of hidden nodes. Our analysis here suggests that these later layers might serve mostly in a variance reducing capacity. The full network's test accuracy is higher than any of its components, which is possible only since their mistakes have relatively low correlation.

## 3  Empirical Results

In this section we will discuss the results from a large scale classification study comparing the performance of a deep neural network and a random forest on a collection of 116 data sets from the UCI Machine Learning Repository. We also discuss empirical ensemble decompositions for a number of trained neural networks from binary classification tasks.

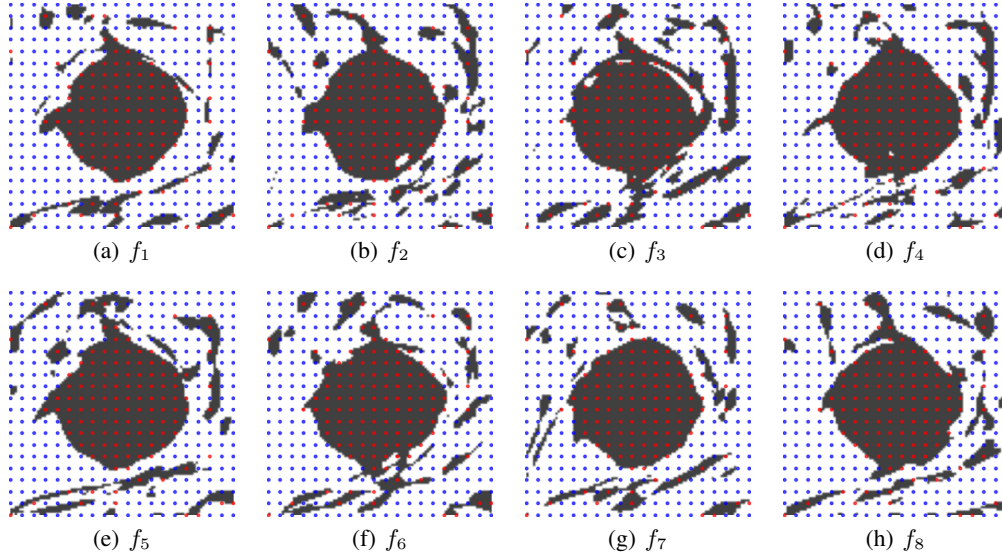

|  (a) $f_1$  |  (b) $f_2$  |  (c) $f_3$  |  (d) $f_4$  |
|  (e) $f_5$  |  (f) $f_6$  |  (g) $f_7$  |  (h) $f_8$  |

Figure 2: Decision surfaces implied from a decomposition of the neural network from Section 2.3.

## 3.1 Data Summary

The collection of data sets we consider were first analyzed in a large-scale study comparing the accuracy of 147 different classifiers [10]. This collection is salient for several reasons. First, Fernández-Delgado et al. [10] found that random forests had the best accuracy of all the classifiers in the study (neural networks with many layers were not included). Thus, random forests can be considered a gold standard to which compare competing classifiers. Second, this collection of data sets presents a very different test bed from the usual image and speech data sets found in the neural network literature. In particular, these data sets span a wide variety of domains, including agriculture, credit scoring, health outcomes, ecology, and engineering applications, to name a few. Importantly, these data sets also reflect a number of realities found in data analysis in areas apart from computer science, such as highly imbalanced classes, non-trivial Bayes error rates, and discrete (categorical) features.

These data sets tend to have a small number of observations: the median number of training cases is 601, and the smallest data set has only 10 observations. It is interesting to note that these small sizes lead to highly overparameterized models: on average, each network as 155 parameters per training observation. The number of features ranges from 3 to 262, and half of data sets include categorical features. Finally, the number of classes ranges from 2 to 100. See Table 1 for a more detailed summary of the data sets.

|  | CATEGORICAL | CLASSES | FEATURES | N |
|---|---|---|---|---|
| MIN | 0 | 2 | 3 | 10 |
| 25% | 0 | 2 | 8 | 208 |
| 50% | 4 | 3 | 15 | 601 |
| 75% | 8 | 6 | 33 | 2201 |
| MAX | 256 | 100 | 262 | 67557 |

Table 1: Dataset Summary

## 3.2 Experimental Setting

For each data set in our collection, we fit three classifiers: a random forest, and neural networks with and without dropout. Importantly, the training process was completely non-adaptive. One of

our goals was to illustrate the extent to which deep neural networks can be used as "off-the-shelf" classifiers.

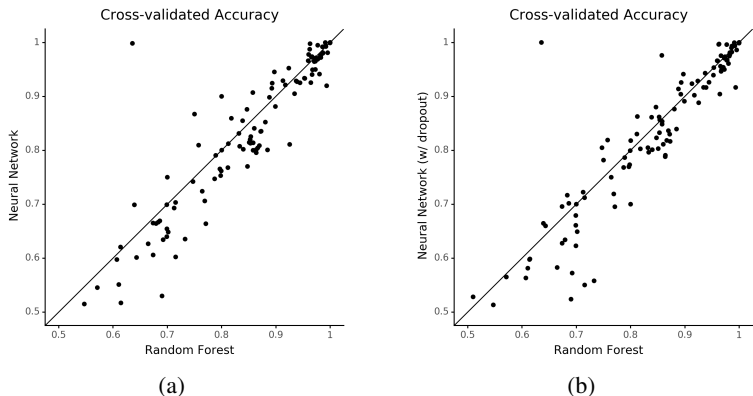

Figure 3: Plots of cross-validated accuracy. Each point corresponds to a data set.

### 3.2.1   Implementation Details

Both networks shared the following architecture and training specifications:

- 10 hidden layers
- 100 nodes per layer
- 200 epochs of gradient descent using Adam optimizer with a learning rate of $0.001$ [15].
- He-initialization for each hidden layer [12]
- Elu activation function [8].

Our choice of architecture was chosen simply to ensure that each network had the capacity to achieve perfect training accuracy in most cases. In practice, we found that networks without dropout achieved 100% training accuracy after a couple dozen epochs of training.

The only difference between the networks involved the presence of explicit regularization. More specifically, one network was fit using dropout with a keep-rate of 0.85, while the other network was fit without explicit regularization. Dropout can be thought of as a ridge-type penalty is often used to mitigate over-fitting [21]. There are other types of regularization not considered in this paper, including weight decay, early stopping, and max-norm constraints.

Each random forest was fit with 500 trees, using defaults known to work well in practice [6]. In particular, in each training instance, the number of randomly chosen splits to consider at each tree node was $\sqrt{p}$, where $p$ is the number of input variables. Although we did not tune this parameter, the performance we observe is very similar to that found in [10].

We turn first to Figure 3, which plots the cross-validated accuracy of the neural network classifiers and the random forest for each data set. In the first figure, we see that a random forest outperforms an unregularized neural network on 72 out of 116 data sets, although by a small margin. The mean difference in accuracy is $2.4\%$, which is statistically significant at the $0.01$ level by a Wilcoxon signed rank test. We notice that the gap between the two classifiers tends to be the smallest on data sets with low Bayes error rate - those points in the upper right hand portion of the plot. We also notice that there exists data sets for which either a random forest or a neural network significantly outperforms the other. For example, a neural network achieves an accuracy of 90.3% on the *monks-2* data set, compared to 62.9% for a random forest. Incidentally, the base-rate for the majority class is 65.0% percent in this data set, indicating that the random forest has completely overfit the data.

Turning to the second plot in Figure 3, we see that dropout improves the performance of the neural network relative to the random forest. The mean difference between classifiers is now decreased to 1.5%, which is still significant at the 0.01 level. The largest improvement in accuracy occurs in data

sets for which the random forest achieved an accuracy of between 75% and 85%. It is also worth noting that the performance difference between the neural networks with and without dropout is less than one percent, and this difference is not statistically significant.

While it might not be surprising that explicit regularization helps when fitting noisy data sets, it *is* surprising that its absence does not lead to a complete collapse in performance. Neural networks with many layers are dramatically more expressive than shallow networks [3], which suggests deeper networks should also be more susceptible to fitting noise when the Bayes error rate is non-zero. We find this is not the case.

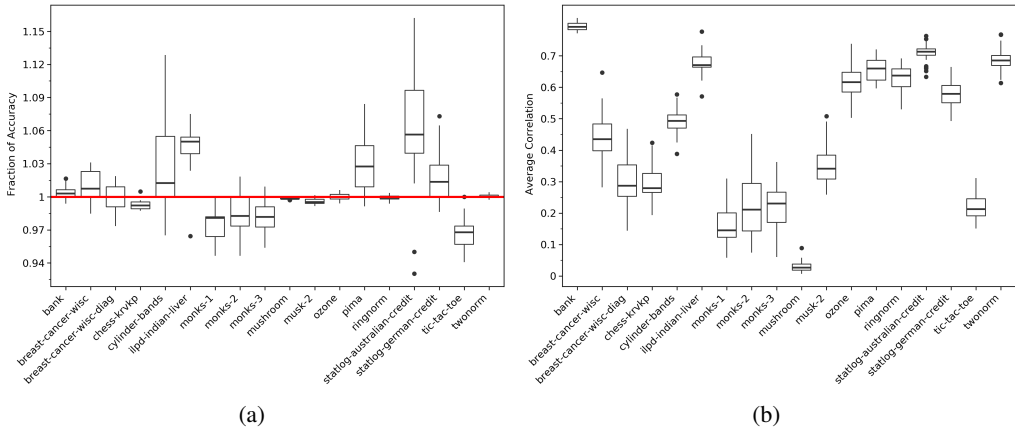

(a)                                                         (b)

Figure 4: The left figure displays the ratio of test error of the best sub-network to the full network, while the right figure displays the average error correlation among sub-networks.

## 3.3   Ensemble Formation

We will now carry over the ensemble decomposition analysis from Section 2 to the binary classifiers fit in Section 3 using $K = 10$ sub-networks. We restrict our analysis to data sets with at least 500 observations, and for which the fitted neural network achieved 100% training accuracy. All results are reported over 25 randomly chosen 80-20 training-testing splits, and all metrics we report were obtained from the testing split.

In the first figure of Figure 4, we report the test accuracy of the best sub-network as a fraction of the full network. For example, in the *statlog-australian-credit* data set, the average value of this fraction was 1.06 (over all 25 training-testing splits), meaning that the best sub-network outperformed the full network by 6% on average. Conversely, in other data sets, such as *tic-tac-toe* data set, the best sub-network had worse performance than the full network across all training-testing splits.

In the second figure of Figure 4, we report the error correlation averaged over the 10 sub-networks. Ensembles of classifiers work best when the mistakes made by each component have low correlation - this is the precise motivation for the random forest algorithm. Strikingly, we observe that the errors made by the sub-networks tend to have low correlation in every data set. Empirically, this illustrates that a neural network can be decomposed as a collection of *diverse* sub-networks. In particular, the error correlation in the *tic-tac-toe* data set is around 0.25, which reconciles our observation that the full network performed better than the best sub-network.

## 4   Discussion

We established that deep neural networks can generalize well on small, noisy data sets despite memorizing the training data. In order to explain this behavior, we offered a novel perspective on neural networks which views them through the lens of ensemble classifiers. Some commonly used wisdom when training neural networks is to choose an architecture which allows one sufficient capacity to fit the training data, and then scale back with regularization [2]. Contrast this with the mantra of a random forest: fit the training data perfectly with very deep decision trees, and then rely

on randomization and averaging for variance reduction [1]. We have shown that the same mantra can be applied to a deep neural network. Rather than each layer presenting an ever increasing hierarchy of features, it is plausible that the final layers offer an ensemble mechanism.

Finally, we remark that the small size of data sets we consider and relatively small network sizes have obvious computational advantages, which allow for rapid experimentation. Some of the recent norms proposed for explaining neural network generalization are intractable on networks with millions of parameters: Schatten norms, for example, require computing a full SVD [11]. In the settings we consider here, such calculations are trivial. Future research should aim to discern a mechanism for the decorrelation we observe, and to explore the link between decorrelation and generalization.

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
