[Reviews · NeurIPS 2018]

Reviewer 1



This paper presents an interesting idea, which is that deep neural networks are able to maintain reasonable generalization performance, even on relatively small datasets, because they can be viewed as an ensemble of uncorrelated sub-networks. Quality: The decomposition method seems reasonable, except for the requirement for the model and the sub-nets to achieve 100% training accuracy. While there are some datasets where this will be reasonable (often high-dimensional datasets), there are others where such an approach would work very badly. That seems to me a fundamental weakness of the approach, especially if there are datasets of that nature where deep neural nets still perform reasonably well. For a random forest, we have an unweighted combination of base classifiers, but it is a learned combination in the case of the decomposed sub-networks, and the weights are tuned on the training data. Unless each component has similar weights it is not clear to me the comparison with RF is truly apposite. The top hat dataset in figure 1 does not seem greatly convincing as the generalization of RF and the deep neural network looks quite poor, both have over-fitted the training data quite badly (hard to see how it could have over-fitted the data any more that it has done). It isn't clear how e.g. categorical attributes have been coded. This should be set out clearly so the work is replicable. The simplicity of the experimental set-up, however, with the key parameters stated, is a nice aspect of the paper. I'm not sure I agree with the description of the results in lines 224-239. In both cases, the average difference in performance seems quite large, and indeed the deep neural networks are statistically inferior to the RF, suggesting they have not performed all that well, although it is a good observation that their performance has not completely collapsed, that seems a rather weak finding. The thing that seems missing to me is a discussion of whether the correlation in the sub-nets is correlated with their generalisation performance. While there is a plot of the error correlation in Fig 4 (b) there doesn't seem to be a way of relating that to generalisation performance. Perhaps re-order the datasets according to the ratio of their performance relative to RF? It is good to see deep learning applied to smaller datasets, rather than the image datasets typically used as benchmarks, it is just as interesting to find out what doesn't work, as well as what does. Clarity: The paper is quite well written, I had no problems understanding it (I hope!). Rao should be capitalised in reference 16. Originality: The idea seems reasonable and novel, I don't recall having seen a similar decomposition suggested. Significance: I think this paper is interesting. I have found MLP networks with large hidden layer sizes often perform reasonably well on small datasets, and it would be useful to have some insight into why this happens (although with a single hidden layer, the decomposition into de-correlated sub-networks seems less applicable). In my experience deep neural networks do not work as well as "shallow" neural networks for this type of data, but even for datasets where there are better algorithms, such insight is still useful.

Reviewer 2



The aim of this paper is to show that deep neural networks can generalize well on small datasets, contrarily as what is accepted in the community. To do that, deep neural networks are presented/interpreted as ensembles of poor classifiers and this is achieved creatively by restricting the power of representation of the subnetworks. The paper is well organized and presented. Mathematics is sound and clear and the experiments are well chosen, including a toy model problem. The results of the restricted neural networks (with and without dropout) are compared to a random forest classifier. The paper can be a very interesting contribution to the conference and can be accepted as is.

Reviewer 3



My main concern is the novelty of the work if we look at it from an ensemble learning perspective. I understand the main point of the work is to examine a 'new' way of decomposing an existing DNN to understand why it works. Unfortunately I don't think the paper has really answered that question. My point is (deep) neural network can be flexible enough to simulate most of the current ensembling strategies/topologies, so it can be designed to overfit or underfit on these small datasets based on what we want it to do. Random forests use bagging (as regularization) to find a reasonable good diversities between overfitted trees in order to get good prediction performance. Same, if DNN could somehow balance overfitted subnets and do a smart ensemble at the final layers, it will reduce the chance of overfitting of the whole network. It is still bias and variance reduction I think, so can we really claim DNN can generalise on small dataset? Probably not. I believe this is a solid work to confirm that if proper DNN topologies are used, deep neural networks don't have to overfit small datasets.